# Analysis of the Peritumoral Tissue Unveils Cellular Changes Associated with a High Risk of Recurrence

**DOI:** 10.3390/cancers15133450

**Published:** 2023-06-30

**Authors:** Audrey Michot, Pauline Lagarde, Tom Lesluyes, Elodie Darbo, Agnès Neuville, Jessica Baud, Gaëlle Perot, Iris Bonomo, Mathilde Maire, Maxime Michot, Jean-Michel Coindre, François Le Loarer, Frédéric Chibon

**Affiliations:** 1Bordeaux Institute of Oncology, BRIC U1312, INSERM, 33000 Bordeaux, France; elodie.darbo@u-bordeaux.fr (E.D.); jessica.massiere@u-bordeaux.fr (J.B.); f.le-loarer@bordeaux.unicancer.fr (F.L.L.); 2Institut Bergonié, Centre de Lutte Contre le Cancer de Bordeaux, 33076 Bordeaux, France; pauline.ruggiero.lagarde@gmail.com (P.L.); tom.lesluyes@crick.ac.uk (T.L.); a.neuville@anapath.fr (A.N.); i.bonomo@bordeaux.unicancer.fr (I.B.); m.maire@bordeaux.unicancer.fr (M.M.); maxime.michot0502@gmail.com (M.M.); j.coindre@bordeaux.unicancer.fr (J.-M.C.); 3Department of Biopathology, Bergonié Institute, Université Victor Segalen Site Carreire, Bordeaux 2, 33076 Bordeaux, France; 4OncoSarc, INSERM U1037, Cancer Research Center in Toulouse (CRCT), 31000 Toulouse, France; gaelle.perot@inserm.fr (G.P.); frederic.chibon@inserm.fr (F.C.); 5Department of Pathology, Institut Claudius Régaud, IUCT-Oncopole, 31000 Toulouse, France

**Keywords:** soft-tissue sarcoma, molecular changes, recurrence, peritumoral tissue, peritumoral capsule

## Abstract

**Simple Summary:**

Recurrence in soft-tissue sarcomas represents a major drawback in the therapeutic management of patients. It may lead to amputation in cases of extreme STS or even to a therapeutical cul-de-sac in other localizations. The molecular identification of patients that are likely to develop a recurrence would represent a major breakthrough in adapting treatment and monitoring patients at risk. In this project, we identified two distinct cellular profiles of peritumoral tissue associated with different clinical behavior. This characterization could help clinicians to tailor neoadjuvant treatments based on a given patient’s risk.

**Abstract:**

Background: The management of soft-tissue sarcoma (STS) relies on a multidisciplinary approach involving specialized oncological surgery combined with other adjuvant therapies to achieve optimal local disease control. Purpose and Results: Genomic and transcriptomic pseudocapsules of 20 prospective sarcomas were analyzed and revealed to be correlated with a higher risk of recurrence after surgery. Conclusions: A peritumoral environment that has been remodeled and infiltrated by M2 macrophages, and is less expressive of healthy tissue, would pose a significant risk of relapse and require more aggressive treatment strategies.

## 1. Introduction

Soft-tissue sarcomas (STS) are malignant tumors often located in the limbs showing varied mesenchymal differentiations and clinical presentations [1,2]. Their management is based on a multidisciplinary approach. Neoadjuvant therapies, depending on tumor FNCLCC grade, are used to control local disease [3,4]. If possible, oncological surgery with limb-sparing is then performed. Currently, clinicians need more precise prognostic biomarkers to tailor the different use of neoadjuvant treatments based on the patient’s risk evolution but also to perform conservative surgery in low-risk patients. A molecular grading “CINSARC” was developed several years ago [5] and is currently being evaluated in clinical trials (CIRSARC NCT03805022; NEOSARCOMICS NCT02789384). A local risk recurrence of 10% is estimated at 5 years after complete excision surgery [6,7,8]. Local recurrences constitute a major threat in sarcoma patients, eventually leading to amputation or death if radical surgery cannot be performed. In contrast, overall survival (OS) is correlated with one’s metastatic status and depends on several factors [9]. The major prognostic factors are the patient’s age, the tumor size, the tumor depth, the histological tumor grade (assessed with the FNCLCC scheme), the locoregional involvement, the initial surgery, and the tumor margins status [3,10,11]. However, it is crucial to define new biomarkers associated with local or distant recurrence to better differentiate high-risk and low-risk patients and subsequently tailor treatments for patients. Interestingly, in the 1980s, Enneking et al. described a “reactive zone” surrounding the tumor, which is also referred to as the peritumoral “capsule” [12]. This area is typically removed by surgical oncologists to improve local control, although little is known about its contribution to disease control. This area is variably composed of a mixture of tumor cells and inflammatory elements susceptible to favor recurrence if the resection passes through it [13]. Nevertheless, the contributions of this reactive zone or the healthy tissue adjacent to the tumor-on-tumor promotion, progression, and recurrence remain largely unknown in STS. So far, biological studies have mainly focused on parameters within the tumor core, while the determinants of the margins have been poorly analyzed. Molecular profiling of the reactive capsule might therefore inform margin status and help to identify markers associated with local recurrence. Indeed, although this area is not mainly composed of tumor cells, it may contain a small subset of infiltrating tumor cells embedded in remodeled fibrous tissue that is often rich in inflammatory cells and could be prognostic as in other cancers [14,15,16,17]. These tissue modifications are poorly understood and often obscured by the fact that studies usually focus primarily on the tumor bulk. Studies have highlighted the capacity of tumor cells to remodel the surrounding tissue via chemokines and growth factors secreted by recruited inflammatory cells and/or cancer-associated fibroblasts (CAFs). These adjoining cells would participate in the aggressiveness of cancer cells and their ability to remodel adjacent tissue [18]. Importantly, there is no consensual definition of what pseudocapsule tissue is made of; for instance, does the presence of tumor cells evidenced in microscopy preclude us from calling it a true pseudocapsule but rather incipient tumor infiltration? Local recurrence may occur through three main mechanisms: the transformation of pre-cancerous cells, the reseeding of the area by persistent tumor cells, or an indirect remodeling area. First, some studies have hypothesized that the adjacent peritumoral tissue might contain a primary clone of pre-tumor cells that may subsequently transform and be at the origin of a new tumor [19]. In carcinomas, the concept of “field of cancerization” is well-known and could likewise apply to soft tissue tumors that arise from mesenchymal cells, although the exact cell of origin of sarcomas remains unknown in most sarcoma types [20,21]. So far, molecular data are lacking in mesenchymal tissue. Second, the peritumoral tissue may contain infiltrating tumor cells that may later reseed the area. This model has been evidenced in cerebral tumors [22,23,24,25,26,27]. In sarcomas, this hypothesis is supported by evidence of sarcoma cells within the peritumoral “edema” that is akin to the “pseudocapsule” seen in human tumors visualized on imaging (MRI) and present at a distance from the tumor itself [18]. The study of the composition of the pseudocapsule in sarcomas may identify adjoining cells (i.e., CAF and inflammatory cells) whose presence may correlate with a higher risk of recurrence. Finally, a third hypothesis might be considered through indirect remodeling via secreted signals, chemokines and growth factors secreted by recruited inflammatory cells, and/or cancer-associated fibroblasts (CAFs). The importance of this interaction might reflect the tumor’s aggressiveness. To determine the influence of these different factors, we characterized the pseudocapsule of sarcoma patients at genomic and transcriptomic levels in a prospective series of 20 sarcomas, evidencing markers that correlate with a higher risk of recurrence after surgery.

## 2. Materials and Methods

### 2.1. Collection of Samples and Patient Information

This prospective study included a series of 20 successive adult patients operated on for primary STS of the limbs or trunk walls at the Institut Bergonié between April 2013 and January 2014. All patients included in the study underwent surgery at Institut Bergonié and provided their consent to be featured in this study. This study was approved by the ethical committee of our institution.

### 2.2. Pathological Assessment

All surgical specimens were studied fresh and after formalin fixation. Three areas were sampled for the purpose of the study defined as follows and as shown in Figure 1:The healthy tissue located more than 1 cm away from the tumor is referred to as HT.The tissue that comes into direct contact with the tumor and is free of tumor at the macroscopic level and upon microscopic examination is considered an R1 margin. The term R1 is used to describe this tissue. Additionally, the composition of the capsule was observed.The tissue taken from within the tumor mass is referred to as T.

Microscopically, all tumor specimens were analyzed by two specialized pathologists, and sarcomas were classified according to the latest edition of the World Health Organization’s (WHO) classification [9]. Each area was sampled fresh to collect both frozen tissue and a formalin-fixed paraffin-embedded mirror block. Finally, we performed a mirror analysis on the tissues at the intersection between T and R1, searching for tumor cells in both tissues. All R1 blocks were confirmed to be devoid of tumor cells following microscopic examination. No tumor cells were identified in HT- and R1-selected areas.

### 2.3. Nucleic Acid Isolation

Deoxyribonucleic acid (DNA) was isolated from snap-frozen samples for array-based comparative genomic hybridization (CGH). Genomic DNA was isolated with a standard phenol-chloroform extraction protocol after RNase treatment. Ribonucleic acid (RNA) was extracted for gene expression studies. Total RNAs were extracted from frozen samples with Trizol reagent^®^ (Gibco BRL, Life Technologies, Carlsbad, CA, USA) and purified with the RNeasy Min Elute Clean-up Kit^®^ (Qiagen, Courtaboeuf, France) according to the manufacturer’s procedure. RNA quality was checked on an Agilent 2100 bioanalyzer^®^ (Agilent Technologies, Massy, France) according to the manufacturer’s recommendations.

### 2.4. Array CGH

DNA was hybridized to 8 × 60 K whole-genome Agilent arrays (G4450A) according to the manufacturer’s protocol. The ADM-2 algorithm of Agilent Genomic Workbench Lite Edition 6.5.0.18 was used to identify DNA copy number anomalies at the probe level. A gain of copy number was defined as a log 2 ratio > 0.25, and a loss of copy number was defined as a log 2 ratio < −0.25.

### 2.5. Gene Expression

Gene expression analysis was carried out using Agilent Whole human 4 × 44 K Genome Oligo Array^®^ (Agilent Technologies) according to the manufacturer’s protocol. All microarrays were simultaneously normalized using the quantile algorithm. *t*-tests were performed using Genespring (Agilent Technologies), and *p*-values were adjusted using the Benjamini–Hochberg procedure. The *p*-value and Fold change cut-off for gene selection were 0.05 and 2, respectively. Gene ontology (GO) analysis was performed to establish statistical enrichment in GO terms using Gorilla (Gene Ontology enrichment analysis and visualization tool). Matched supervised (per case) and unsupervised clustering on the entire cohort were performed.

### 2.6. Statistical Analysis

Computations and plots, external to Genespring, were performed using R (v3.1.1). In gene expression, from the 41,093 probes in Agilent chips, we removed both controls (93) and probes not associated with known genes (10,064). Then, when multiple probes matched a given gene, we computed the interquartile range and kept the highest one for each single gene. In the end, 19,595 genes (i.e., probes) were used for statistical analyses (e.g., Kruskal–Wallis, pairwise Wilcoxon with Benjamini–Hochberg correction for multiple tests).

### 2.7. CIBERSORT

CIBERSORT is an analytical tool developed by Newman et al. [28] to provide an estimation of the abundance of immune cell types using gene expression data. It is a computational method for quantifying cell fractions from bulk tissue gene expression profiles. All samples are tested, and profiling of inflammatory cells is noted. The value of differential expression of immune cells such as macrophages M2, macrophages M0, monocytes, mast cells resting, B cells memory, T cells CD4 naïve, neutrophils, T cells follicular helper, macrophages M1, T regulators, Natural killer cells resting, T cells CD4 memory activated, dendritic cells resting, dendritic cells activated, T cells gamma delta, B cells naïve, eosinophils, mast cells activated, T cells CD4 memory resting, T cells CD8, NK cells activated was noticed for each sample. CIBERSORT was applied according to the method outlined by Newman et al.

### 2.8. Immunohistochemistry Stainings

Immunohistochemistry was performed on the automated Ventana Discovery XT staining platform (Ventana Medical Systems, Oro Valley, AZ, USA). In brief, FFPE slides were deparaffinized, and antigen retrieval was performed by heat-induced epitope retrieval using standard CC1 reagent (Tris-based buffer, pH 8.0; Ventana Medical Systems). The slides were incubated with antibodies directed against cMAF (EPR16484, Abcam, Cambridge, UK, 1/250) and CD68 (PG-M1, Dako, Nowy Sącz, Poland, 1/100) and then with OmniMap HRP-conjugated anti-rabbit or anti-mouse IgG (Ventana; Roche, Basel, Switzerland), respectively. The presence of bound antibodies was revealed by tyramide signal amplification using DISCOVERY Purple or 3,3′-diaminobenzidine (DAB) chromogen detection kit (Roche) for IHC. The slides were finally counterstained with hematoxylin (Roche), cover-slipped, and digitized using a multispectral imaging platform (PhenoImager HT, Akoya, Menlo Park, CA, USA).

## 3. Results

Twenty soft tissue sarcoma patients were analyzed, the characteristics of which are listed in Table 1. The mean age of the patients was 62 years (range 44 to 85). The mean tumor size was 80 mm (range 25 to 170). Most tumors were located on the limbs (85%, n = 17/20); most were high-grade and deep-seated (Table 1). Most tumors (17/20 = 85%) were pleomorphic sarcomas with highly a rearranged genome: 5 leiomyosarcomas (LMS), 4 myxofibrosarcomas (MFS), 7 undifferentiated pleomorphic sarcomas (UPS), 2 liposarcomas (LPS), 1 pleomorphic rhabdomyosarcoma (RMS), and 1 low grade fibro myxoid sarcoma (LGFM). The mean follow-up was 57 months (range 10–118), 12 patients had a relapse (60%), including 3 local recurrences (15%) and 10 metastases (50%), and 13 patients died of disease (65%). One patient had simultaneous local recurrence and metastasis (counted as metastasis, which is the major event).

The genome of all samples was profiled via array CGH. Tumor genomes showed numerous, mostly non-recurrent rearrangements (Appendix A). In contrast, the genomic profiles of the healthy (HT) and surrounding tissues (R1) were all diploid and did not exhibit copy number alteration (Figure 2 and Appendix A). HT and R1 areas were composed of fibrous and/or inflammatory tissue and were devoid of tumor cells (confirmed upon histological analysis of the mirror blocks). The expression profiles of HT and R1 areas were compared, identifying 1345 genes significantly differentially expressed between the two areas. In R1 relative to HT, 868 genes were upregulated, being composed of genes involved in the synthesis of the extracellular matrix (ECM); extracellular matrix remodeling; immune system; inflammatory response; response to stimulus; and regulation of cell migration, cell motility, or adhesion (Table 2). In R1 relative to HT, the 477 downregulated genes are involved in the muscle system process and oxidative metabolism (Table 2).

Then, after identifying the R1 entity, we set out to determine whether we could recategorize our healthy samples of HT and R1.

### 3.1. Comparative Transcriptomic Analysis of R1 Areas

The expression profiles of all areas (HT, R1, and T) of all samples (n = 60) were compared by unsupervised paired hierarchical clustering analysis. The profiles clustered in three main clusters are represented in Figure 3. The first group contained all 20 tumor samples (T areas). The second was composed of 15 HT and 8 R1, including 8 paired. Finally, the third cluster was composed of 12 R1 and 5 HT. All these samples were taken from different sarcoma subtypes, including the LMS (all LMS of this series), two MFS, two UPS, one LPS, and one RMS. The R1 samples were divided into two classes: those clustered with healthy tissue samples (HT) are referred to as the R1.h samples, while the others closer to tumor tissue are referred to as the R1.t samples. The expression profiles of R1.h versus R1.t were compared. Compared to R1.t, the R1.h samples overexpressed 682 genes involved in muscular system and oxidative metabolism. In contrast, the R1.t samples were enriched in 120 genes involved in the synthesis of ECM, metabolism of ECM, lipids and steroids, locomotion, cell motility, and immune system process. A gradient of expression of genes involved in the muscle system was found between R1.t, R1.h, and HT. These muscular genes are the most differentially expressed between R1.h vs. R1.t. R1.h presented an expression in genes involved in the muscular system at the same level in the surrounding healthy tissue and in the distant tissue (example in Figure 4).

### 3.2. Assessment of GI in Tumor Samples

The genomic index (GI) was calculated with the array comparative genomic hybridization (aCGH) profiles of tumors. For stratification, the genomic index was calculated as follows: GI = A2/C, where A corresponded to the total number of alterations (segmental gains or losses), and C corresponded to the number of chromosomes affected by these alterations. GI cutoff (i.e., 10) was chosen in accordance with a previous study [29]. In the first R1.h group, the GI had a mean value of 72 (range 5–192) (Appendix A). In the second R1.t group, the GI had a mean value of 165 (range 9–385), higher than R1.h (Appendix A)

### 3.3. Comparative Histological Analysis of R1.h versus R1.t Groups

Upon microscopic examination, it was observed that the samples of the R1.h cluster were made of normal striated muscle in most cases (67%, n = 6/12) or normal fibrous and adipose tissue in the others shown in Figure 5. The samples of group R1.t contained inflammatory tissue or fibrous adipose tissue and did not contain muscle. The modifications seen in the tissue variably included inflammatory infiltrates. Tumor cells were identified neither in the peritumoral (R1) nor in the distant healthy tissue (HT).

### 3.4. Clinical Significance of Patient’s Stratification According to Their R1 Tissue Nature

In the R1.t group, the patients developed adverse outcomes in a mean time of 19 months, which included 6 events (67%, n = 6/12)—5 metastasis-related (41%, n = 5/12) and 1 local recurrence (11%, n = 1/12)—with 9 deaths (75%, n = 9/12). All R1 and HT areas of the LMS of the series clustered together intermingled with the HT and R1 areas of MFS (n = 2), two UPS, one RMS, and one LPS. In this group, local recurrence was present in one case (11%). In the R1.h group, the mean time for the first event was 39 months, which included four events (50%, n = 4/8)—declines in three metastases (37.5%, n = 3/8), one local recurrence (12.5%, n = 1/8), and one both (12.5%, n = 1/8)—with four deaths (50%, n = 4/8). The difference was not statistically significant between the two groups (*p* = 0.4, *p* = 0.78, *p* = 0.3, respectively). A Kaplan–Meier analysis of overall survival in each group was performed (Figure 6), and the probability of local or metastasis recurrence depending on time was also analyzed (Figure 7).

### 3.5. Comparison of the Immune Reactions between R1.h and R1.t

After applying the CIBERSORT analysis method, four clusters were distinguished (Figure 8, Appendix A). The first cluster, Immune1 (I1, n = 10) in the head of Figure 5 contained eight HT and two R1. This group was rich in terms of monocyte population. The second cluster, Immune 2 (I2, n = 5), included five T samples enriched in M0 and M2 Macrophages. In this group, 75% of patients (6 samples/8) had developed metastasis and died of disease. The largest third cluster, I3, contained 9 T, 14 R1, and 5 HT (enriched only in M2 macrophages and a low level of monocytes). Interestingly, it lumped together all the samples from four patients, of whom three were still alive at follow-up. In the last cluster, I4, there were three T, four R1, and seven HT that displayed a low level of immune infiltration in the different populations.

After comparing the immune profiles of R1.h and R1.t (Table 3 and Table 4), it was observed that the R1.t group was rich in macrophages M2 and M0, although the difference did not reach statistical significance compared to R1.h (0.025 vs. 0.022 and 0.015 vs. 0.004, respectively). There was also a higher level of mast cells, T CD4+ memory lymphocytes, T CD8+ lymphocytes, and activated NK cells in the R1.t group (0.11 vs. 0.078, 0.107 vs. 0.0694, 0.063 vs. 0.057, 0.098 vs. 0.096, respectively). The R1.h group was rich in monocytes, plasma cells, and B cells memory compared to the R1.t samples, although the difference was not statistically significant (0.078 vs. 0.11, 0.09 vs. 0.013, 0.045 vs. 0.0275). To visualize and confirm these results, we performed double M2 macrophage labeling on two cases to distinguish between the R1.t and R1.h groups (Figure 9A,B for patients # 9 and # 18, respectively).

## 4. Discussion

The quality of the surgical margin is an important parameter used to guide the clinical management of sarcoma patients [3,30,31,32]. Nonetheless, local recurrences may occur even after an optimal surgery is deemed complete (referred to as R0 in the clinics) by both the surgeon and, despite the absence of sarcoma infiltration in the surgical margins upon microscopic examination, by the pathologists [3]. The determinants of risk of local recurrence remain poorly studied, and it remains unknown whether they overlap with those of distant relapse and metastasis. However, optimal clinical management in a tertiary care center improves the overall survival of patients [33,34]. Metastatic evolution and local recurrence affect the evolution of patients after complete surgical resection. Most studies have focused on the biological determinants of tumor cells to predict clinical outcomes. In this study, we tried to identify biological parameters associated with the risk of recurrence in the margins and healthy tissue, as some factors may escape the traditional microscopic analysis performed by pathologists. Moreover, it is known that tissues surrounding tumors may contain inflammatory infiltrates reflecting the host’s reaction against the tumor. Indeed, we were not able to detect the copy number imbalances of the tumor in the normal tissue analyzed, confirming their benign nature even in patients that subsequently developed local recurrences. Moreover, the R1 and HT samples were all devoid of cancer cell infiltration upon microscopic analysis, precluding the presence of a minor tumor infiltration not detected by genomic profiling, as reported in models of glioblastoma [22].

Interestingly, we were able to identify significant modifications in the expression profiles between HT and R1 samples, which is in line with the concept of pseudocapsules (remodeled and modified tissue surrounding the tumor). In the surrounding tissue, there is indeed an overexpression of genes involved in extracellular matrix (ECM) synthesis or remodeling. ECM remodeling has been shown to give the tumor cells the ability to invade normal tissue [35]. The R1 zones experienced upregulation in the transcripts involved in the extracellular matrix and structural organization, as well as in genes involved in immune and inflammatory processes. These results confirm that the periphery of the tumor is at the forefront of the interaction of tumor cells with the host. This “pseudocapsule” should reflect the long-term ecological process of the crosstalk between tumor and stroma cells [36]. The direct interaction of cancer cells with the immune cells has been demonstrated in many cancer models and correlated with patient survival [37]. The anticancer immune reaction can kill cancer cells through the action of cytotoxic lymphocytes, but this natural reaction can be dampened by cancer cells [38,39,40]. In this study, we evidenced a downregulation of the expression of genes involved in the muscle system process and contraction and in oxidation-reduction process in the R1 samples compared to HT samples.

It is unknown at which level the surrounding tissue may contribute to the development and maintenance of primary tumors or may account for subsequent local recurrences. Similar data have been shown in other cancers, including lung, head and neck, breast, and colorectal carcinomas [41]. These studies suggested that molecular modifications may occur in the surrounding tissue even before the development of cancer and may play a role in the initiation and progression of the disease. Two distinct types of R1 samples at the expression level in our study that tended to correlate with distinct clinical outcomes were identified. The importance of this interaction might reflect the tumor’s aggressiveness [42]. This role of reciprocal communication with tumor cells has been shown to be pivotal in the development and progression of epithelial carcinoma [42].

The R1.h group was indeed associated with an upregulation of 682 genes involved in the muscular system and oxidative metabolism compared to R1.t. This particular aspect was more similar to HT. In fact, some HT samples that were clustered upon transcription analysis with R1 samples were referred to as “R1-like HT samples.” Compared to the R1.t group, these samples evidenced a downregulation of genes involved in the synthesis and metabolism of the extracellular matrix, lipids and steroids, locomotion, cell motility, and immune system process. This consideration is also consistent with a trend of a higher level of monocytes, plasma cells, and B cells memory. On a histological level, these samples mostly contained normal striated muscle without visible reaction or tumor infiltration. In the R1.h group, the GI of the corresponding tumors amounted to a mean value of 72 and correlated with a trend of longer time in the occurrence of adverse events (mean time of 39 months; not significant). A high genomic rearranged tumor was associated with a more aggressive R1 (not significant). In comparison, the samples belonging to the R1.t group mostly contained rearranged or inflammatory fibrous and adipose tissue. These samples were more frequently infiltrated by inflammatory cells upon microscopy, and there was an enrichment in macrophages that became visible during the CIBERSORT analysis. This trend of more infiltration of macrophages M2 and M0 is associated with a shorter time to occurrence, with a mean time of 19 months, and a more rearranged tumor (in terms of the level of genomic alterations).

Interestingly, other groups have shown that the presence of macrophages M2 in tumors is associated with a poor prognosis due to their immunosuppressive property (18–20). The R1.t group was associated with a higher level of mast cells, T CD4+ lymphocytes memory, T CD8+ lymphocytes, and activated NK, in keeping with the microscopic analysis of the samples on the slide as found also in osteosarcoma [43]. This R1.t group, which seems to include patients with more aggressively adverse outcomes, presents an immune response with lineage T and a more inflammatory environment at the histological level. Thanks to Mellman et al., it is known that a cytotoxic reaction with CD8 + and NK cells triggers cancer cell death and may secondarily amplify the antitumor immune response [37].

These cellular changes were not known until then, and they require further study on a larger cohort with longer clinical follow-up (in order to consider events). Our study suggests that prognostic factors might be inferred from the study of the peritumor capsule.

## 5. Conclusions

In conclusion, we have highlighted that certain biological properties of the peritumoral tissue can provide information about the patient’s risk of recurrence. A peritumoral area that has been remodeled, infiltrated by M2 macrophages, identified by IHC, and is low in the expression of healthy tissue may be at a greater risk of relapse. Identifying patients who are at risk of recurrence can help to tailor their management.

## Figures and Tables

**Figure 1 cancers-15-03450-f001:**
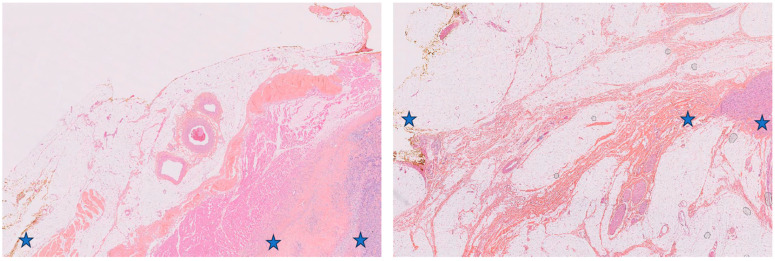
Pathological changes observed by HE staining from HT to R1 to T. (Case 5 and 14, respectively).

**Figure 2 cancers-15-03450-f002:**
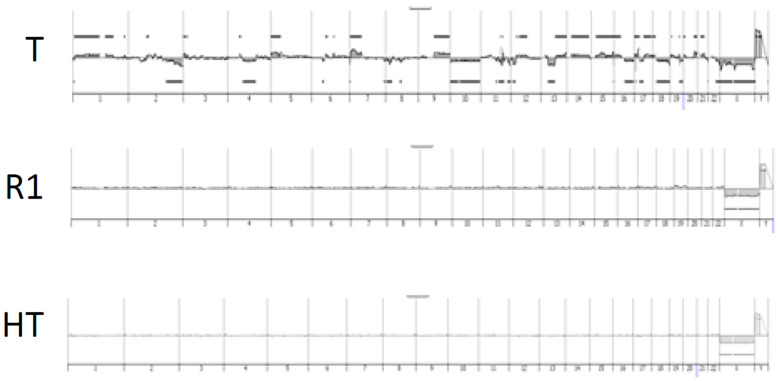
Genomic profiles obtained via CGH on three areas: T tumor, R1 non-tumoral surrounding tissue, and HT healthy tissue at distance. Tumor genome was highly rearranged, while R1 and HT were flat.

**Figure 3 cancers-15-03450-f003:**
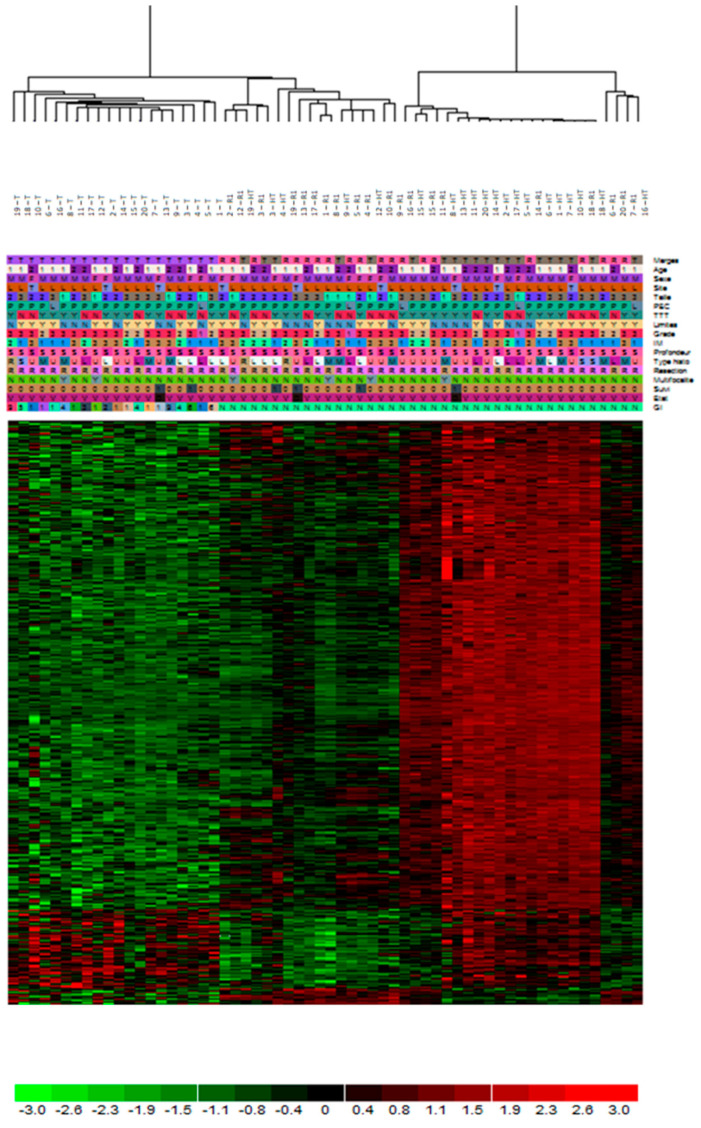
Unsupervised non-hierarchical paired clustering with all probes of all entities for all samples. Three groups are designed according to their expression differential.

**Figure 4 cancers-15-03450-f004:**
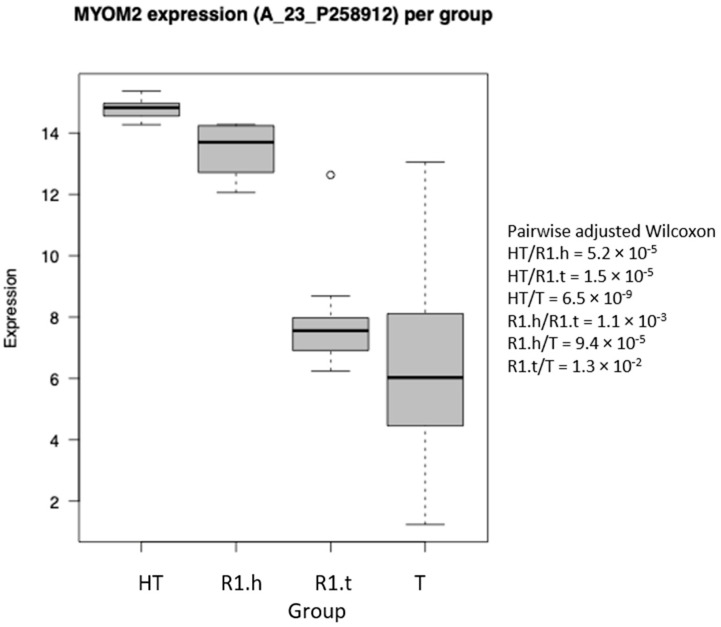
Difference in the expression of the gene MYOM2 from GO:0003012 in our four groups (HT, R1.h, R1.t, and T).

**Figure 5 cancers-15-03450-f005:**
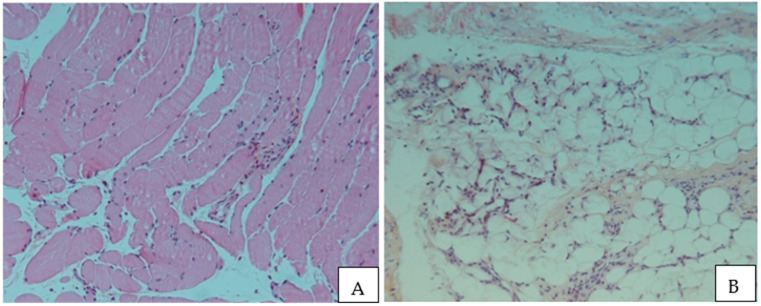
(**A**): Normal striated muscle; (**B**): Rearranged fibroadipose tissue.

**Figure 6 cancers-15-03450-f006:**
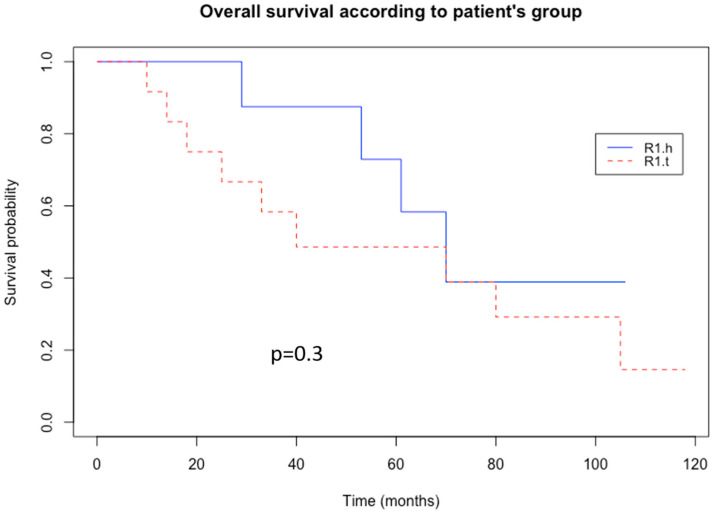
Kaplan–Meier curve of overall survival. R1.h in blue; R1.t in red.

**Figure 7 cancers-15-03450-f007:**
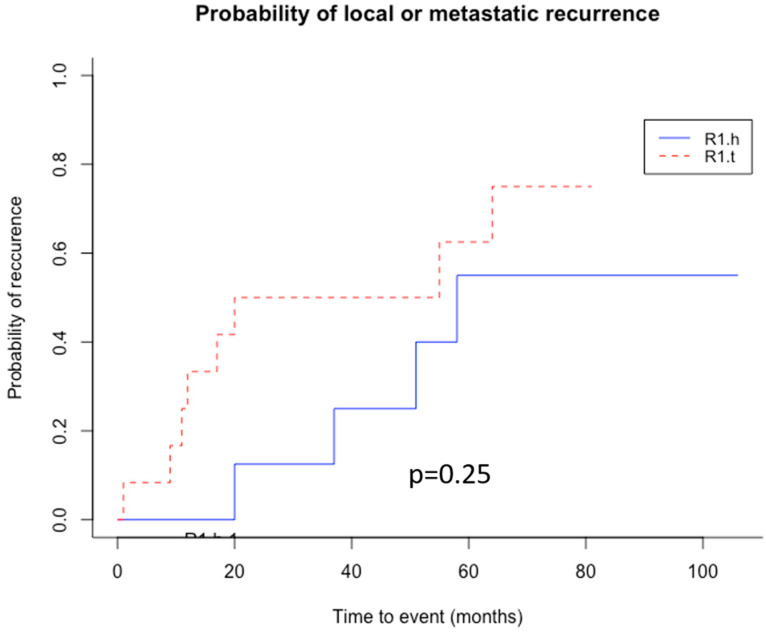
Kaplan–Meier curve for probability of local or metastatic recurrence. R1.h in blue; R1.t in red.

**Figure 8 cancers-15-03450-f008:**
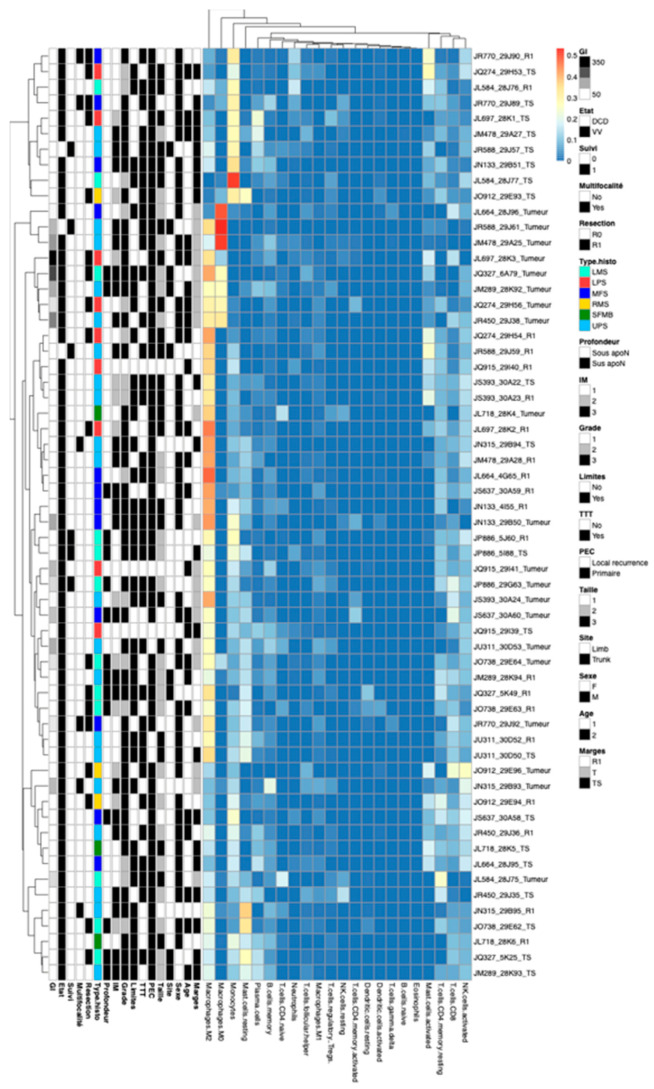
CIBERSORT method: signature of an immune infiltrate of each sample. It was reported the differential expression of immune cells of macrophages M2, macrophages M0, monocytes, mast cells resting, B cells memory, T cells CD4 naïve, neutrophils, T cells follicular helper, macrophages M1, T regulators, Natural killer cells resting, T cells CD4 memory activated, dendritic cells resting, dendritic cells activated, T cells gamma delta, B cells naïve, eosinophils, mast cells activated, T cells CD4 memory resting, T cells CD8, NK cells activated in each area of samples.

**Figure 9 cancers-15-03450-f009:**
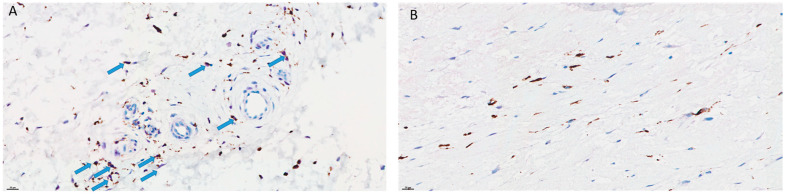
(**A**): Patient #9: numerous cells showing a double marquage for CD-68 and c-MAF highlighting the presence of M2 macrophages. This patient died with metastases 25 months after the diagnosis. (**B**): Patient #18: simple marquage for CD-68 with no M2 cells. This patient is in complete remission 44 months after diagnosis.

**Table 1 cancers-15-03450-t001:** Patients and tumor characteristics: age in years; tumor size in mm; Chemo: chemotherapy; Contouring: tumor margins; LMS: leiomyosarcoma; MFS: myxofibrosarcoma; UPS: undifferentiated sarcoma; LPS: dedifferentiated liposarcoma; LFMBS: low-grade fibromyxoid sarcoma; RMS: rhabdomyosarcoma; CR: complete remission; M: metastasis; follow-up in months; NED: no evolutive disease; AWD: alive with disease; DOD: dead of disease. Identification of the R1 entity.

Case ID	Age (yo)	Size (mm)	Location	Tumor Grade	Extension	Type	Margin Status	Evolution	Time to Event (mo)	Follow Up (mo)	Genomic Index	Vital Status
1	61	100	Limb	2	Deep	LMS	R0	M	55	105	61	DOD
2	53	80	Trunk	3	Supf	LMS	R0	M	12	14	277	DOD
3	85	55	Limb	3	Supf	LMS	R1	CR		80	49	Dead
4	62	70	Limb	3	Supf	LMS	R0	M	1	118	65	AWD
5	72	11	Limb	1	Deep	LMS	R0	CR		36	109	NED
6	57	80	Limb	2	Deep	MFS	R0	CR		97	17	NED
7	46	130	Limb	3	Deep	MFS	R0	M	37	70	138	DOD
8	59	25	Limb	3	Deep	MFS	R1	LR	17	33	41	DOD
9	66	25	Limb	3	Supf	MFS	R0	M	20	25	97	DOD
10	71	65	Trunk	3	Supf	UPS	R0	M	11	18	134	DOD
11	65	75	Limb	3	Deep	UPS	R0	LRM	58	61	158	DOD
12	54	25	Limb	3	Deep	UPS	R0	CR		81	110	NED
13	59	170	Trunk	3	Deep	UPS	R0	M	9	10	114	DOD
14	75	60	Limb	3	Deep	UPS	R0	CR		65	192	NED
15	58	100	Limb	2	Deep	UPS	R0	M	20	29	16	DOD
16	60	110	Limb	3	Deep	UPS	R0	CR		106	127	NED
17	76	150	Limb	3	Deep	LPS	R1	M	64	70	385	DOD
18	44	100	Limb	3	Deep	LGFMS	R0	CR		44	5	NED
19	52	80	Limb	3	Deep	RMS	R1	CR		40	9	Dead
20	75	100	Limb	2	Deep	LPS	R1	LR	51	53	49	DOD
abbreviations:	Age in years	Size in mm		Grade FNCLCC		Type of sarcoma (cf abbreviations)	R0 complete microscopic exerese R1 in contact at microscopic level	M metastasis CR complete remission LR local recurrence	Time in months	Follow up in months		DOD dead of disease; Dead of other cause; NED no evolutive disease; AWD alive with disease

**Table 2 cancers-15-03450-t002:** GO corresponds with upregulated and downregulated genes in R1 relative to HT.

GO Term	Description	*p*-Value	FDR q-Value	Enrichment (N, B, n, b)	Corresponding Most Genes Significant
GO:0030198	extracellular matrix organization	4.76 × 10^-29^	5.91 × 10^-25^	4.28 (16,751, 356, 868, 79)	CYP1B1—cytochrome p450, family 1, subfamily b, polypeptide 1 SFRP2—secreted frizzled-related protein 2 ADAMTS18—adam metallopeptidase with thrombospondin type 1 motif, 18 FBN1—fibrillin 1 BGN—biglycan
GO:0043062	extracellular structure organization	5.82 × 10^-29^	3.61 × 10^-25^	4.27 (16,751, 357, 868, 79)	CYP1B1—cytochrome p450, family 1, subfamily b, polypeptide 1 SFRP2—secreted frizzled-related protein 2 ADAMTS18—adam metallopeptidase with thrombospondin type 1 motif, 18 FBN1—fibrillin 1 BGN—biglycan
GO:0002376	immune system process	5.44 × 10^-19^	2.25 × 10^-15^	2.01 (16,751, 1586, 868, 165)	CFH—complement factor h LRMP—lymphoid-restricted membrane protein CFB—complement factor b WDFY4—wdfy family member 4 IRF8—interferon regulatory factor 8
GO:0006954	inflammatory response	8.22 × 10^-15^	6.81 × 10^-12^	3.12 (16,751, 359, 868, 58)	HFE—hemochromatosis FOLR2—folate receptor 2 (fetal) IL1R1—interleukin 1 receptor, type i CXCR6—chemokine (c-x-c motif) receptor 6 GPR68—g protein-coupled receptor 68
GO:0003012	muscle system process	5.5 × 10^-14^	6.82 × 10^-10^	4.99 (16,751, 225, 477, 32)	SNTA1—syntrophin, alpha 1 SLC6A8—solute carrier family 6 (neurotransmitter transporter), member 8 MAP2K6—mitogen-activated protein kinase kinase 6 DTNA—dystrobrevin, alpha HEY2—hairy/enhancer-of-split related with yrpw motif 2
GO:0006936	muscle contraction	5.89 × 10^-13^	3.66 × 10^-9^	5.26 (16,751, 187, 477, 28)	SNTA1—syntrophin, alpha 1 MYOM1—myomesin 1 SCN7A—sodium channel, voltage-gated, type vii, alpha subunit SLC6A8—solute carrier family 6 (neurotransmitter transporter), member 8 MAP2K6—mitogen-activated protein kinase kinase 6
GO:0055114	oxidation-reduction process	3.98 × 10^-8^	4.95 × 10^-5^	2.17 (16,751, 908, 477, 56)	ACAT1—acetyl-coa acetyltransferase 1 PGM2L1—phosphoglucomutase 2-like 1 NDUFS7—nadh dehydrogenase (ubiquinone) fe-s protein 7, 20kda (nadh-coenzyme q reductase) LYRM7—lyr motif containing 7 COQ9—coenzyme q9 homolog (s. cerevisiae)

**Table 3 cancers-15-03450-t003:** Correspondence between our cluster of DGE and immune data of CIBERSORT in the R1.h group.

R1.t	M2	M0	Monocytes	Mast Cells Resting	Plasma Cells	B Cells Memory	T Cells CD4 Naive	Neutrophils	T Cells Follicular Helper	M1	Tregs	NK Cells Resting	T Cells CD4 Memory Activated	Dendritic Cells Resting	Dendritic Cells Activated	T Cells Gamma Delta	B Cells Naive	Eosinophils	Mast Cells Activated	T Cells CD4 Memory Resting	T Cells CD8	NK Cells Activated
2_R1	0.342	0	0.0141	0.17	0.0051	0.045	0	0.003	0	0.01	0.03	0	0	0.1	0	0	0	0	0	0.156	0.071	0.06
12_R1	0.234	0.034	0.056	0.37	0	0.085	0.05	0	0.015	0.01	0.01	0	0	0	0	0	0	0	0	0.016	0.057	0.06
19_HT	0.016	0.083	0.3111	0.26	0.0013	0.014	0	0.001	0.005	0	0.04	0	0	0	0	0	0	0	0.039	0.112	0.047	0.04
3_R1	0.249	0	0.113	0.18	0.0055	0	0.06	0	0.031	0	0	0	0	0.1	0.1	0	0	0	0	0.084	0.106	0.03
3_HT	0.227	0	0.0503	0.33	0.0205	0.047	0	0	0.025	0.02	0	0	0	0	0	0	0	0	0	0.113	0.071	0.06
4_HT	0.222	0	0.234	0.16	0.0041	0.013	0	0.064	0.008	0.01	0	0	0	0	0	0	0	0	0	0.116	0.053	0.11
19_R1	0.098	0	0.1881	0.03	0.0514	0.043	0	0	0	0.01	0	0	0	0	0	0	0	0	0.153	0.193	0.067	0.16
13_R1	0.364	0	0.1186	0	0.0015	0.007	0	0.002	0.015	0.05	0	0	0	0	0	0	0	0	0.247	0.099	0.017	0.08
17_R1	0.457	0	0.064	0.02	0.0101	0.026	0	0	0.005	0.03	0	0	0	0	0	0	0	0	0.085	0.117	0.058	0.13
1_R1	0.16	0.07	0.2793	0	0.0056	0.002	0	0.163	0	0	0	0	0	0	0	0	0	0	0.176	0.102	0.01	0.03
8_R1	0.072	0.014	0.3202	0	0.0157	0.005	0.01	0.128	0.013	0	0.01	0	0	0	0	0	0	0	0.245	0.015	0.007	0.13
9_HT	0.098	0	0.2478	0	0.0101	0.021	0	0.041	0	0.04	0	0	0	0	0	0	0	0	0.09	0.161	0.114	0.18
5_R1	0.366	0	0.1563	0	0	0.019	0	0.049	0.002	0.01	0	0	0	0	0	0	0	0	0.152	0.106	0.028	0.11
4_R1	0.285	0	0.2719	0.01	0.012	0.001	0.01	0	0.001	0.04	0	0	0	0	0	0	0	0	0.013	0.094	0.09	0.16
12_HT	0.424	0	0.1053	0.12	0.0317	0.039	0	0	0	0	0	0	0	0	0	0	0	0	0	0.072	0.083	0.12
10_R1	0.217	0.02	0.082	0.14	0.0419	0.064	0	0.006	0	0.05	0	0	0	0	0	0	0	0	0	0.145	0.129	0.1
9_R1	0.427	0.034	0.0464	0.09	0.0065	0.037	0	0	0.003	0.06	0	0	0	0	0	0	0	0	0	0.122	0.065	0.11

**Table 4 cancers-15-03450-t004:** Correspondence between our cluster of DGE and immune data of CIBERSORT in the R1.t group.

R1 h	M2	M0	Monocytes	Mast Cells Resting	Plasma Cells	B Cells Memory	T Cells CD4 Naive	Neutrophils	T Cells Follicular Helper	M1	Tregs	NK Cells Resting	T Cells CD4 Memory Activated	Dendritic Cells Resting	Dendritic Cells Activated	T Cells Gamma Delta	B Cells Naive	Eosinophils	Mast Cells Activated	T Cells CD4 Memory Resting	T Cells CD8	NK Cells Activated
16_R1	0.283	0	0.1129	0.2	0.0807	0.039	0	0	0	0.02	0	0	0	0	0	0	0	0	0	0.063	0.119	0.09
15_HT	0.319	0	0.0744	0.05	0.0533	0.008	0	0	0.016	0.03	0	0	0	0	0	0	0	0	0.131	0.09	0.078	0.16
15_R1	0.33	0	0.0892	0	0.0291	0.03	0	0	0	0.02	0	0	0	0	0.01	0	0	0	0.209	0.083	0.064	0.14
11_R1	0.417	0	0.0597	0.12	0.0546	0	0	0	0.013	0.02	0	0	0	0	0	0	0	0	0.033	0.078	0.046	0.15
8_HT	0.036	0.072	0.3012	0	0.0563	0.125	0	0.068	0	0	0	0.07	0	0	0.01	0	0	0	0.097	0.105	0.049	0.02
11_HT	0.11	0	0.2757	0.07	0.2012	0	0	0	0.076	0	0	0.02	0	0	0	0	0.01	0	0.034	0.088	0.034	0.08
13_HT	0.187	0	0.2923	0	0.1376	0.041	0.05	0.034	0.023	0	0.01	0	0	0	0	0	0	0	0.12	0	0.013	0.09
20_HT	0.053	0	0.2066	0	0.0202	0.011	0	0.099	0.04	0.04	0	0	0	0	0.01	0	0	0	0.276	0.06	0.039	0.15
14_HT	0.153	0	0.0865	0.15	0.1158	0	0	0	0.068	0.04	0.05	0.15	0	0	0	0	0	0	0	0.163	0.021	0
2_HT	0.167	0	0.1487	0.27	0.1484	0.046	0	0	0.012	0.01	0.01	0	0	0	0	0	0	0	0	0	0.103	0.08
17_HT	0.094	0	0.2735	0.01	0.2342	0.019	0	0.042	0.028	0	0.09	0.05	0	0	0.01	0	0	0	0	0.055	0.053	0.04
5_HT	0.249	0	0.1298	0.08	0.1158	0.094	0.05	0.03	0.013	0.01	0.01	0	0	0	0	0	0	0	0	0.037	0.066	0.12
14_R1	0.202	0	0.1965	0.03	0.107	0.036	0	0	0.015	0	0.02	0	0	0	0.01	0	0	0	0.091	0.061	0.075	0.16
6_HT	0.088	0	0.169	0	0.1531	0.055	0	0.027	0.033	0.03	0	0	0	0	0	0	0	0	0.16	0.066	0.049	0.16
1_HT	0	0.024	0.5147	0	0.027	0.053	0.02	0	0.006	0	0.05	0	0	0	0.01	0	0	0	0.087	0.045	0.024	0.13
7_HT	0.164	0	0.3505	0.03	0.1103	0.095	0	0.009	0	0	0	0	0	0	0	0	0	0	0.069	0.109	0.031	0.03
10_HT	0.197	0	0.1116	0.2	0.1466	0.075	0	0	0.033	0.01	0	0	0	0	0	0	0	0	0	0.068	0.067	0.1
18_R1	0.198	0.006	0.2142	0.16	0.1108	0.089	0.02	0	0	0	0.03	0	0	0	0	0	0	0	0	0	0.092	0.08
18_HT	0.152	0	0.1612	0	0.0866	0.059	0	0	0.013	0.01	0.02	0	0	0	0	0	0	0	0.178	0.101	0.087	0.14
6_R1	0.479	0	0.0848	0.1	0.0106	0.058	0	0	0.025	0.01	0.01	0	0	0	0.01	0	0	0	0	0.101	0.03	0.08
20_R1	0.42	0	0.0456	0	0.0224	0.029	0	0.005	0.028	0.02	0	0	0	0	0	0	0	0	0.215	0.074	0.028	0.11
7_R1	0.41	0	0.175	0.12	0.0188	0.058	0.06	0	0	0.01	0.02	0	0	0	0	0	0	0	0	0.114	0	0.03
16_HT	0.354	0	0.1426	0.21	0.0301	0.021	0	0	0.005	0.02	0.01	0	0	0	0	0	0	0	0	0.035	0.114	0.06

## Data Availability

Not applicable.

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
