# Peer review of "Analysis of the Peritumoral Tissue Unveils Cellular Changes Associated with a High Risk of Recurrence"

_cancers, 2023, doi:10.3390/cancers15133450_

Round 1
Reviewer 1 Report
Comments for authors and editorial teams:
The study entitled "Analysis of the peritumoral tissue unveils cellular changes associated with a high risk of recurrence” is novel and interesting. However, some minor concerns should be considered to improve the novelty as well as the flow of the manuscript:
1- Abstract: Acceptable
2- Keywords: Acceptable
3- Introduction: Acceptable but could be better by wider literature review particularly related new & update articles from domestic researchers.
4- Methodology: Acceptable
5- Results: Acceptable
6- Discussion: Could be scientifically than present level using more and wider literature review and explanation of related and domestic survey and of course then using from them as Bibliography/References.
7- Bibliography/References: More domestic and update Ref(s). are necessary. Authors should do more literature review for using related domestic articles and therefore some more domestic Ref(s). (using from domestic Ref(s). is scientific discipline for publication of academic data.)
8- I could not see running title in manuscript. It is compulsory.
9- Table & Graph quality: Acceptable
Final decision: Manuscript is acceptable for publication after minor correction
Important: Improve the quality this nice article by using new references like:
Googe PB, Theocharis S, Pergaris A, Li H, Yan Y, McKenna EF, Moschos SJ
Theragnostic significance of tumor-infiltrating lymphocytes and Ki67 in BRAFV600-mutant metastatic melanoma (BRIM-3 trial).
Current problems in cancer. 2022
Zhou Q, Ke X, Xue C, Li S, Huang X, Zhang B, Zhou J
A Nomogram for Predicting Early Recurrence in Patients with High-Grade Gliomas.
World neurosurgery. 2022
Angelucci C, D'Alessio A, Sorrentino S, Biamonte F, Moscato U, Mangiola A, Sica G, Iacopino F
Immunohistochemical Analysis of DNA Repair- and Drug-Efflux-Associated Molecules in Tumor and Peritumor Areas of Glioblastoma.
International journal of molecular sciences. 2021
Martínez R, Tapia G, De Muga S, Hernández A, Cao MG, Teixidó C, Urrea V, García E, Pedreño-López S, Ibarz L, Blanco J, Clotet B, Cabrera C
Combined assessment of peritumoral Th1/Th2 polarization and peripheral immunity as a new biomarker in the prediction of BCG response in patients with high-risk NMIBC.
Oncoimmunology. 2019
Pehlivanoglu B, Aysal A, Agalar C, Egeli T, Ozbilgin M, Unek T, Oztop I, Sagol O
Peritumoral histopathologic findings in patients with chronic viral hepatitis-associated hepatocellular carcinoma.
APMIS : acta pathologica, microbiologica, et immunologica Scandinavica. 2022
Campbell SC, Muñoz-Ballester C, Chaunsali L, Mills WA, Yang JH, Sontheimer H, Robel S
Potassium and glutamate transport is impaired in scar-forming tumor-associated astrocytes.
Neurochemistry international. 2019
Ahn H, Won Lee J, Jang SH, Ju Lee H, Lee JH, Oh MH, Mi Lee S
Prognostic significance of imaging features of peritumoral adipose tissue in FDG PET/CT of patients with colorectal cancer.
European journal of radiology. 2021
Shen Z, Xiao J, Wang J, Lu L, Wan X, Cai X
Peritumoral ductular reaction can be a prognostic factor for intrahepatic cholangiocarcinoma.
BMC gastroenterology. 2020
Scalas G, Parmeggiani A, Martella C, Tuzzato G, Bianchi G, Facchini G, Clinca R, Spinnato P
Magnetic resonance imaging of soft tissue sarcoma: features related to prognosis.
European journal of orthopaedic surgery & traumatology : orthopedie traumatologie. 2021
Wadiura LI, Millesi M, Makolli J, Wais J, Kiesel B, Mischkulnig M, Mercea PA, Roetzer T, Knosp E, Rössler K, Widhalm G
High Diagnostic Accuracy of Visible 5-ALA Fluorescence in Meningioma Surgery According to Histopathological Analysis of Tumor Bulk and Peritumoral Tissue.
Lasers in surgery and medicine. 2020
Frati A, Armocida D, Arcidiacono UA, Pesce A, D'Andrea G, Cofano F, Garbossa D, Santoro A
Peritumoral Brain Edema in Relation to Tumor Size Is a Variable That Influences the Risk of Recurrence in Intracranial Meningiomas.
Tomography (Ann Arbor, Mich.). 2022
Zhang F, Wang Y, Chen G, Li Z, Xing X, Putz-Bankuti C, Stauber RE, Liu X, Madl T
Growing Human Hepatocellular Tumors Undergo a Global Metabolic Reprogramming.
Cancers. 2021
Furuse M, Kuwabara H, Ikeda N, Hattori Y, Ichikawa T, Kagawa N, Kikuta K, Tamai S, Nakada M, Wakabayashi T, Wanibuchi M, Kuroiwa T, Hirose Y, Miyatake SI
PD-L1 and PD-L2 expression in the tumor microenvironment including peritumoral tissue in primary central nervous system lymphoma.
BMC cancer. 2020
Wei H, Jiang H, Zheng T, Zhang Z, Yang C, Ye Z, Duan T, Song B
LI-RADS category 5 hepatocellular carcinoma: preoperative gadoxetic acid-enhanced MRI for early recurrence risk stratification after curative resection.
European radiology. 2020
He HL, Wang Q, Liu L, Luo NB, Su DK, Jin GQ
Peritumoral edema in preoperative magnetic resonance imaging is an independent prognostic factor for hepatocellular carcinoma.
Clinical imaging. 2021
Guo X, Zhu Y, Wang X, Xu K, Hong Y
Peritumoral Edema Is Associated With Postoperative Hemorrhage and Reoperation Following Vestibular Schwannoma Surgery.
Frontiers in oncology. 2021
Chiu FY, Le NQK, Chen CY
A Multiparametric MRI-Based Radiomics Analysis to Efficiently Classify Tumor Subregions of Glioblastoma: A Pilot Study in Machine Learning.
Journal of clinical medicine. 2021
Qiu J, Deng K, Wang P, Chen C, Luo Y, Yuan S, Wen J
Application of diffusion kurtosis imaging to the study of edema in solid and peritumoral areas of glioma.
Magnetic resonance imaging. 2021
Muscas G, van Niftrik CHB, Sebök M, Seystahl K, Piccirelli M, Stippich C, Weller M, Regli L, Fierstra J
Hemodynamic investigation of peritumoral impaired blood oxygenation-level dependent cerebrovascular reactivity in patients with diffuse glioma.
Magnetic resonance imaging. 2020
Hsu JB, Lee TY, Cheng SJ, Lee GA, Chen YC, Le NQK, Huang SW, Kuo DP, Li YT, Chang TH, Chen CY
Identification of Differentially Expressed Genes in Different Glioblastoma Regions and Their Association with Cancer Stem Cell Development and Temozolomide Response.
Journal of personalized medicine. 2021
Yang Y, Wang MC, Tian T, Huang J, Yuan SX, Liu L, Zhu P, Gu FM, Fu SY, Jiang BG, Liu FC, Pan ZY, Zhou WP
A High Preoperative Platelet-Lymphocyte Ratio Is a Negative Predictor of Survival After Liver Resection for Hepatitis B Virus-Related Hepatocellular Carcinoma: A Retrospective Study.
Frontiers in oncology. 2020
Lee S, Kang TW, Song KD, Lee MW, Rhim H, Lim HK, Kim SY, Sinn DH, Kim JM, Kim K, Ha SY
Effect of Microvascular Invasion Risk on Early Recurrence of Hepatocellular Carcinoma After Surgery and Radiofrequency Ablation.
Annals of surgery. 2021
Xu L, Wan Y, Luo C, Yang J, Yang P, Chen F, Wang J, Niu T
Integrating intratumoral and peritumoral features to predict tumor recurrence in intrahepatic cholangiocarcinoma.
Physics in medicine and biology. 2021
Flammang I, Reese M, Yang Z, Eble JA, Dhayat SA
Tumor-Suppressive miR-192-5p Has Prognostic Value in Pancreatic Ductal Adenocarcinoma.
Cancers. 2020
Beebe E, Pöschel A, Kunz L, Wolski W, Motamed Z, Meier D, Guscetti F, Nolff MC, Markkanen E
Proteomic profiling of canine fibrosarcoma and adjacent peritumoral tissue.
Neoplasia (New York, N.Y.). 2022
Azab MA, Ghozy S, Hassanein SF, Azzam AY
Specific Preoperative Dynamic Contrast-Enhanced MRI Semi-quantitative Markers Can Correlate With Vascularity in Specific Areas of Glioblastoma Tissue and Predict Recurrence.
Cureus. 2021
Cui Y, Zeng W, Jiang H, Ren X, Lin S, Fan Y, Liu Y, Zhao J
Higher Cho/NAA Ratio in Postoperative Peritumoral Edema Zone Is Associated With Earlier Recurrence of Glioblastoma.
Frontiers in neurology. 2020
Wang LL, Li JF, Lei JQ, Guo SL, Li JK, Xu YS, Dou Y
The value of the signal intensity of peritumoral tissue on Gd-EOB-DTPA dynamic enhanced MRI in assessment of microvascular invasion and pathological grade of hepatocellular carcinoma.
Medicine. 2021
Vasilescu AM, Andriesi Rusu DF, Bradea C, Vlad N, Lupascu-Ursulescu C, Cianga Spiridon IA, Trofin AM, Tarcoveanu E, Lupascu CD
Protective or Risk Factors for Postoperative Pancreatic Fistulas in Malignant Pathology.
Life (Basel, Switzerland). 2021
Paliashvili K, Popov A, Kalber TL, Patrick PS, Hayes A, Henley A, Raynaud FI, Ahmed HU, Day RM
Peritumoral Delivery of Docetaxel-TIPS Microparticles for Prostate Cancer Adjuvant Therapy.
Advanced therapeutics. 2020
Varghese JR, Gurusamy DS, Kalyanasundaram S, Kalyanaraman S
Role of podoplanin, E-cadherin, Ki-67 in the dissemination of tumor cells in ovarian surface epithelial carcinoma-An immunohistochemical study.
Annals of diagnostic pathology. 2022
Silva M, Vivancos C, Duffau H
The Concept of «Peritumoral Zone» in Diffuse Low-Grade Gliomas: Oncological and Functional Implications for a Connectome-Guided Therapeutic Attitude.
Brain sciences. 2022
Author Response
The study entitled "Analysis of the peritumoral tissue unveils cellular changes associated with a high risk of recurrence” is novel and interesting. However, some minor concerns should be considered to improve the novelty as well as the flow of the manuscript:
1- Abstract: Acceptable
2- Keywords: Acceptable
3- Introduction: Acceptable but could be better by wider literature review particularly related new & update articles from domestic researchers.
We would like to thank the reviewer and we modified the introduction with adding more references.
4- Methodology: Acceptable
5- Results: Acceptable
6- Discussion: Could be scientifically than present level using more and wider literature review and explanation of related and domestic survey and of course then using from them as Bibliography/References.
It has been done accordingly to the proposed references.
7- Bibliography/References: More domestic and update Ref(s). are necessary. Authors should do more literature review for using related domestic articles and therefore some more domestic Ref(s). (using from domestic Ref(s). is scientific discipline for publication of academic data.)
We added more references.
8- I could not see running title in manuscript. It is compulsory.
9- Table & Graph quality: Acceptable
We propose a running title as “Impact of peritumoral tissue in soft-tissue-sarcoma progression" if necessary.
Reviewer 2 Report
Τhe title reflects the study. This manuscript presents a clear and clinically useful message. It is well written in terms of clarity, style, and use of English language. Materials and methods approach are sufficiently detailed. The discussion section explains adequately the subject in the context of published information. The conclusions accurately and clearly explain the main clinical message. The length is ideal. All figures are of good quality and relevant to the clinical message. All references are appropriate and current.
Author Response
Reviewer 2:
Τhe title reflects the study. This manuscript presents a clear and clinically useful message. It is well written in terms of clarity, style, and use of English language. Materials and methods approach are sufficiently detailed. The discussion section explains adequately the subject in the context of published information. The conclusions accurately and clearly explain the main clinical message. The length is ideal. All figures are of good quality and relevant to the clinical message. All references are appropriate and current.
We would like to thank the reviewer for this consideration.
Reviewer 3 Report
An interesting study to show that genomic and transcriptomic pseudocapsules of prospective sarcomas were analyzed and revealed to be correlated with a higher risk of recurrence after surgery. Some point should be noted as below.
1) The title should be better as “Analysis of the peritumoral tissue unveils cellular changes associated with a high risk of recurrence in soft-tissue sarcoma”.
2) Instead of Figure 1 “Cross-section of a tumor”, it is better to show the pathological changes (changes observed by HE staining ) in these groups such as Tumor, R1-t, R1-h and HT group.
3) Since genes changes including M2 were examined to be expressed differently in these groups, it needs to be confirmed by IHC method or other methods in these groups.
4) In Table 2, some selected UP and DOWN genes should be better listed.
5) The description in some parts of this article is confusing. For example, the “S” in Figure 2; The “T S” in Figure 3; Line 234, others shown in “Figures 5”? it might be Figure 4? The authors need to carefully check all the parts in this paper.
6) It is interesting that such as “ECM remodeling”, “The direct interaction of cancer cells with the immune cells” were mentioned in this study. Indeed, cancer cells and TME is considered as a pathological ecosystem, and pathological changes during the development of human diseases including cancer should be thought as “Ecological pathology” (Theranostics 2023; 13(5):1607-1631. https://www.thno.org/v13p1607.htm). Based on this concept, the “pseudocapsule” should reflect the long-term ecological process of the crosstalk between tumor and stromal cells. Such advanced view might be helpful for this paper.
Author Response
Reviewer 3:
An interesting study to show that genomic and transcriptomic pseudocapsules of prospective sarcomas were analyzed and revealed to be correlated with a higher risk of recurrence after surgery. Some point should be noted as below.
1) The title should be better as “Analysis of the peritumoral tissue unveils cellular changes associated with a high risk of recurrence in soft-tissue sarcoma”.
We would like to thank the reviewer and we changed the title with adding “in soft-tissue
sarcoma."
2) Instead of Figure 1 “Cross-section of a tumor”, it is better to show the pathological changes (changes observed by HE staining ) in these groups such as Tumor, R1-t, R1-h and HT group.
We have replaced the photograph of the tumor specimen by a histological slide as described above.
3) Since genes changes including M2 were examined to be expressed differently in these groups, it needs to be confirmed by IHC method or other methods in these groups.
We have added an IHC complentary analysis with M2 examination in Figures 9.
4) In Table 2, some selected UP and DOWN genes should be better listed.
We have replaced the GO alone by GO associated with significant genes involved.
5) The description in some parts of this article is confusing. For example, the “S” in Figure 2; The “T S” in Figure 3; Line 234, others shown in “Figures 5”? it might be Figure 4? The authors need to carefully check all the parts in this paper.
Figure 2 was modified with HT, Figure 3 also with T, R1 and HT and in the text, we modified the line 234, « shown in Figure 4 ».
6) It is interesting that such as “ECM remodeling”, “The direct interaction of cancer cells with the immune cells” were mentioned in this study. Indeed, cancer cells and TME is considered as a pathological ecosystem, and pathological changes during the development of human diseases including cancer should be thought as “Ecological pathology” (Theranostics 2023; 13(5):1607-1631. https://www.thno.org/v13p1607.htm). Based on this concept, the “pseudocapsule” should reflect the long-term ecological process of the crosstalk between tumor and stromal cells. Such advanced view might be helpful for this paper.
It has been done and referred in the discussion.
Round 2
Reviewer 3 Report
The authors have made good modifications according to the review suggestions.